# Antimicrobial Susceptibility of Bacteria Isolated from Freshwater Mussels in the Wildcat Creek Watershed, Indiana, United States

**DOI:** 10.3390/antibiotics12040728

**Published:** 2023-04-08

**Authors:** John E. Ekakoro, Lynn F. Guptill, G. Kenitra Hendrix, Lauren Dorsey, Audrey Ruple

**Affiliations:** 1Department of Public and Ecosystem Health, College of Veterinary Medicine, Cornell University, Ithaca, NY 14853, USA; 2Department of Veterinary Clinical Sciences, College of Veterinary Medicine, Purdue University, West Lafayette, IN 47907, USA; 3Department of Comparative Pathobiology/Indiana Animal Disease Diagnostic Laboratory, College of Veterinary Medicine, Purdue University, West Lafayette, IN 47907, USA; 4Department of Population Health Sciences, Virginia-Maryland College of Veterinary Medicine, Virginia Tech, Blacksburg, VA 24061, USA

**Keywords:** antimicrobial resistance, bacterial culture, antimicrobial susceptibility testing, freshwater mussels

## Abstract

Antimicrobial resistance (AMR) is a global health crisis that threatens the health of humans and animals. The spread of resistance among species may occur through our shared environment. Prevention of AMR requires integrated monitoring systems, and these systems must account for the presence of AMR in the environment in order to be effective. The purpose of this study was to establish and pilot a set of procedures for utilizing freshwater mussels as a means of surveillance for microbes with AMR in Indiana waterways. One hundred and eighty freshwater mussels were sampled from three sites along the Wildcat Creek watershed in north-central Indiana. Specimens were evaluated for the presence of ESKAPE pathogens (*Enterococcus faecium*, *Staphylococcus aureus*, *Klebsiella pneumoniae*, *Acinetobacter baumannii*, *Pseudomonas aeruginosa*, *Enterobacter* species), *Escherichia coli*, *Campylobacter*, and *Salmonella* species, and the isolates were tested for antimicrobial resistance. A total of 24 bacterial isolates were obtained from tissue homogenates of freshwater mussels collected at a site directly downstream from Kokomo, Indiana. Of these, 17 were *Enterobacter* spp., five were *Escherichia coli*, one was *Pseudomonas aeruginosa*, and one was *Klebsiella pneumoniae*. All isolates were resistant to three or more antimicrobial drug classes. Further work is necessary to determine the source of the bacterial species found in the mussels.

## 1. Introduction

The emergence of antimicrobial resistance (AMR) is a global crisis threatening the health of humans and animals [1,2]. Infections caused by antimicrobial-resistant bacteria are generally more difficult to treat, tend to recur and lead to significant morbidity and mortality [3]. A robust understanding of the challenge of AMR through rigorous research within the One Health framework is necessary for the mitigation of this challenge. Antimicrobial resistance is an ancient and natural phenomenon [4]. However, the non-judicious use of antimicrobials in humans, animals, and agriculture is a known driver of AMR [5,6]. Worldwide, the occurrence of AMR-associated infections is increasing, and the World Health Organization (WHO) included AMR as one of the top ten threats to global health in 2019 [7,8]. The WHO has called for AMR surveillance systems to be established in order to help combat the spread of AMR, but gaps in surveillance remain as most established systems utilize laboratory data derived from clinical samples collected from humans and animals [9].

Water is a known pathway for the spread of AMR in the environment [6], and elements such as wastewater and wildlife interactions can introduce AMR in surface waters [10]. Globally, contamination of waterways with AMR bacteria and AMR genes results from wastewater effluents from hospitals, pharmaceutical industries, crop agriculture, aquaculture, and animal production [11]. Sewage is a known reservoir of AMR genes [12], and inadequate treatment of wastewater at treatment plants contributes to the spread of AMR [13]. Environmental sources for AMR can serve as a reservoir for the transfer of AMR to humans and animals. Currently, environmental surveillance for AMR continues to lag behind other efforts aimed at mitigating the spread of AMR [14]. Human exposure to AMR bacteria in the environment can occur in recreational areas, e.g., recreational water, through consumption of raw or undercooked fresh produce grown in contaminated fields, consumption of raw or undercooked shellfish, through drinking water, through urban water, e.g., fountains, and via ambient air [15]. Thus, surveillance for AMR organisms in the environment should be considered an integral component of a One Health AMR surveillance system [14,16].

Environmental sampling for AMR organisms can be challenging for several reasons, including time and expense [17]. Freshwater mussels are known to perform important functions in aquatic ecosystems and act as bio-monitors of environmental pollution, water quality, toxin levels, bacteria, and viruses in aquatic systems [18,19,20,21]. Although diverse bacterial species that appear to be natural flora of mussels have been isolated from several mussel species, there is no adequate evidence showing these bacteria are commensals or pathogenic in mussels [22]. Mussels are omnivores that feed on bacteria, algae, and detritus and play important functions in food webs [23]. They also accumulate contaminants of public health importance when they filter water [24]. Thus, sampling mussels rather than sampling water sources directly may be a more efficient way to monitor AMR in waterways.

Results from a study conducted in Sweden showed that zebra mussels (*Dreissena polymorpha*) are suitable for bio-monitoring for fecal inland water contamination and could be used as sentinels for AMR [25]. A study conducted in New Zealand found resistance to one or more antimicrobials in the majority of bacteria isolated from freshwater mussels [26]. Another study conducted in Portugal found high resistance to mainly β-lactams in bacteria isolated from mussels, suggesting possible exposure of the mussels to AMR bacteria in their habitat [27]. A study conducted in Germany found a high number of AMR bacteria in mussels when compared to the water, although the AMR bacteria in the sampled water were more diverse [28]. However, the role of freshwater mussels as bio-indicators for AMR in U.S. waterways has not previously been reported.

The Wildcat Creek in north-central Indiana is known to be an impaired stream with high fecal coliform counts [29]. This creek is considered impaired because it does not meet one or more water quality standards, and it is polluted primarily by *Escherichia coli* [30]. The identified sources of pollution for this stream include row crop agriculture and pastures, urban and rural runoff, manure applied to crop fields, straight pipe discharges, home sewage treatment-system disposal, and combined sewer overflow outlets [30]. Antimicrobial-resistant *Escherichia coli* has been previously detected from various sampling sites in this stream, including a site located directly downstream of Kokomo, an urban human settlement [29]. Until now, bacterial species present in mussels inhabiting the Wildcat Creek watershed had not been isolated and characterized, nor had antimicrobial susceptibility testing (A.S.T.) been performed. The purpose of this pilot study was to determine if freshwater mussels could be utilized as a means of surveillance for AMR organisms in Indiana waterways.

## 2. Results

### 2.1. Bacteria Isolated

No bacteria were isolated from 120 mussels collected at two of the three sampling sites. All the bacterial organisms reported in this paper were isolated from mussels collected from Site A (the map of the sampling sites is provided in the materials and methods section). Of the 60 samples from site A, 20 (33.3%) had growth of a single organism of interest on aerobic culture, two (3.3%) specimens had growth of two distinct Enterobacter sp. isolates (*Enterobacter asburiae* and *Enterobacter cloacae*), each, 12 (20%) specimens did not result in organism identification due to overgrowth on the plates, and no organisms of interest were isolated from 26 (43.3%) specimens. The single organisms isolated from the mussels include *Enterobacter cloacae* (6/24), *Enterobacter asburiae* (11/24), *Escherichia coli* (5/24), *Klebsiella pneumoniae* (1/24), and *Pseudomonas aeruginosa* (1/24). No *Campylobacter* spp. or *Salmonella* spp. were isolated from the samples collected.

Eight (33.3%) of the 24 isolates above were from the plain pocketbook (*Lampsilis cardium*) mussels, and 16 (66.7%) were from the Asian clams (*Corbicula fluminea*). The eight isolates from the plain pocketbooks included five *E. coli*, one *Enterobacter cloacae*, one *Klebsiella pneumoniae*, and one *Pseudomonas aeruginosa*. No bacterial isolates were obtained from the fat mucket mussels (*Lamspilis siliquoidea*).

### 2.2. Antimicrobial Susceptibility

A summary of the antimicrobial susceptibility of the isolates is provided in Table 1. Eleven *Enterobacter asburiae* isolates and five of the Enterobacter cloacae were isolated from the Asian clams. However, antimicrobial susceptibility data for these isolates from the Asian clams were excluded from further analyses due to the possible contamination during the 18 h they were held in the same water bath while still alive after sampling and prior to processing their tissues.

## 3. Discussion

This pilot study found antimicrobial-resistant bacteria in one species of mussels sampled downstream of a human settlement suggesting that mussels can be used for surveillance of AMR in Indiana waterways. A study conducted in Sweden found zebra mussels, another invasive mussel species, to be suitable for monitoring bacterial pathogens in rivers and detecting AMR That Swedish study also suggested that compared to sampling water, sampling mussels may be a more reliable and efficient approach to monitoring AMR in waterways [25].

Several studies have reported the occurrence or increased occurrence of AMR bacteria and AMR genes in the downstream sites of wastewater treatment plants [13,31,32,33,34]. Similar to other studies, the AMR bacterial isolates reported in the present study were detected in one species of mussels collected from only one of three sampling sites located downstream of a human settlement (Kokomo, Indiana), and no bacteria were isolated from all mussels sampled from two of the three sampling sites. The sampling site with bacterial isolates is adjacent to a recreational site where people are allowed to fish or boat and is known to have livestock operations in the watershed [29]. These findings suggest that this waterway may be differentially contaminated with AMR organisms via land use and nearby recreational activities. The Wildcat Creek is also known to be an impaired stream with high *E. coli* and fecal coliform counts [29,30]. It is possible that the findings from this site downstream of a human settlement could be indicative of pollution of the waterways with antimicrobial-resistant bacteria from failing septic tanks, combined with sewer-overflow discharges, discharge of treated wastewater into the stream, animal waste, and stormwater. The absence of bacteria-isolated mussels from two of the three sampling sites provides baseline evidence that could be important in informing future mussel sampling efforts for AMR surveillance on which areas of the creek to concentrate. However, periodic sampling and testing of mussels at each of the three sampling sites could prove useful in discerning temporal patterns in bacterial prevalence and AMR in mussels inhabiting these sites.

Mussel species type and environment determine mussel gut microbiome composition [35]. The gut bacterial microbiome of the Asian clam is known to be diverse and differs with environmental conditions and water bodies [15]. In the present study, no bacteria were isolated from the fat mucket from all three sites. This could be due to differences in the bioaccumulation of bacteria among mussel species. Environmental AMR surveillance efforts could target the utilization of mussel species known to bioaccumulate bacteria and could focus on invasive species. However, there is a need to validate this hypothesis using larger sample sizes. It is possible that the *E. coli*, *Enterobacter cloacae*, and *Klebsiella pneumoniae* isolates reported in this present study are part of the core gut microbiota of the mussels from which they were isolated. This is because a U.S. study that examined the effects of exposure to agricultural contaminants on the microbial community composition of *Lampsilis cardium* suggested that bacteria in the families *Clostridiaceae, Enterobacteriaceae*, and *Rhodobacteraceae* could be part of the core gut microbiota of this mussel species [36].

Although *E. coli* is intrinsically susceptible to almost all clinically relevant antimicrobials, multi-drug-resistant *E. coli* is a major problem observed in both human and veterinary medicine, with resistance often acquired through horizontal gene transfer [37]. As an example, resistance to beta-lactams in *E.coli* of animal and human origin is mediated by acquired genes such as *bla_CTX-M-1_*, *bla_CTX-M-14_*, and *bla_CTX-M-15_* [37]. The *E. coli* isolated in the present study was susceptible to drugs belonging to several medically important antimicrobial classes (sulfonamide, aminoglycoside, beta-lactam, and tetracycline). Medically important antimicrobials are antimicrobial classes used in human medicine [38]. All five *E. coli* isolates were resistant to four antimicrobials (penicillin, tilmicosin, clindamycin, and tiamulin). Fincher and others found a multi-drug resistant *E. coli* O157:H7 isolate in the Wildcat Creek [29]. Although the findings from the present study cannot be directly extrapolated to clinical settings, AMR in clinical settings can be linked to resistance in the environment [39,40].

*Enterobacter* spp. and *Klebsiella* spp. are widespread in the environment and are the causative agents in human infections in the community and in healthcare facilities [41]. Although the organisms identified in this study were susceptible to a number of medically important antimicrobials, resistance to some antimicrobials was observed. Bacterial AMR can result from the inherent properties of the bacterium (intrinsic resistance), through a mutation or through the acquisition of new genetic material from an external source (acquired resistance), and can be induced as a response to a specific signal or environmental situation (adaptive resistance) [3,42]. *Enterobacter* spp. are intrinsically resistant to ampicillin, amoxicillin, amoxicillin–clavulanate, first-generation cephalosporins, and cefoxitin; while *Klebsiella* spp. are intrinsically resistant to penicillins, macrolides, and lincosamides [41,43,44]. The intrinsic resistance to beta-lactams in *Enterobacter* spp. is conferred by the ampC genes [45]. The resistance patterns of the *Enterobacter cloacae* isolated in the present study could be intrinsic. Although the SHV-1 chromosomal beta-lactamase gene is known to confer intrinsic resistance to ampicillin in Klebsiella pneumonia [46], horizontal gene transfer plays a significant role in the acquisition of beta-lactams and quinolone AMR genes by this bacterium [47]. In the present study, the resistance pattern reported for *Klebsiella pneumoniae* could be either intrinsic or acquired. The *Pseudomonas aeruginosa* isolate was susceptible to only aminoglycosides, and the resistance phenotype observed was mainly intrinsic. The lack of susceptibility to penicillin and ceftiofur reported in the present study could be due to a high level of ampC AMR gene expression [48]. *Pseudomonas aeruginosa* is known to be intrinsically resistant to Penicillin G, oxacillin, macrolides, lincosamides, streptogramins, glycopeptides, ampicillin, 1st and 2nd generation cephalosporins, ceftriaxone, tetracyclines, chloramphenicol, and trimethoprim [44].

Similar to all other studies, this present study has limitations. Only three locations in one watershed area were sampled in this study, and water was not sampled for comparison. For some organisms isolated, e.g., *Escherichia coli*, antimicrobial susceptibility profiles for drugs in some antimicrobial classes, e.g., fluoroquinolones, could not be determined using the Bovine BOPO7F AST Plate due to a lack of interpretive criteria. Additionally, the antimicrobial susceptibility of the isolates from the Asian clams was excluded from further analyses due to the possible contamination during pooled storage leading to a reduction in our effective sample size for antimicrobial susceptibility. Despite the limitations, the findings from this pilot study provide evidence that sampling and testing of freshwater mussels could be an important part of AMR environmental surveillance in central Indiana. Our next step is to conduct bacterial culture and antimicrobial susceptibility testing on a larger sample of mussels alongside sediment and water collected from the site where antimicrobial-resistant bacteria were isolated. Additionally, we will use molecular techniques for pathogen identification and assessing the resistome of the isolates.

## 4. Materials and Methods

### 4.1. Test Organisms and Tissue Preparation

A total of 180 freshwater mussels were sampled in June and July 2019 from three sites within the Wildcat Creek watershed located in north-central Indiana (Figure 1). These mussels were evaluated as part of a larger project evaluating mussel health/viability in the ecosystem. The sampled species included the native fat mucket (*Lamspilis siliquoidea*; *n* = 60), the plain pocketbook (*Lampsilis cardium*; *n* = 60), and the non-native Asian clam (*Corbicula fluminea; n* = 60). Twenty mussels of each species were collected from each of the three study sites (Figure 2). At each site, the mussels were all collected in proximity to each other regardless of species. For each mussel, tissue was removed from the shell streamside and placed in a Stomacher^®^ 80 sample bag with 10 mL PBS (0.137 M NaCl, 0.0027 M KCl, 0.0119 M Phosphates, P.H 7.4; Fisher BioReagents, Fisher Scientific, Waltham, MA, USA) and transported to the laboratory in coolers with ice where they were processed within eight hours. Tissues were homogenized for 60 s at high speed using the Stomacher^®^ 80 (Seward Medical, London, UK). The Asian clams collected from site A were held for 18 h in the same water bath while still alive after sampling and prior to processing their tissues.

### 4.2. Bacterial Culture

The bacteriological tests performed included aerobic culture, *Campylobacter* spp. Culture, and *Salmonella* spp. culture. The organisms of interest in aerobic culture included *Enterococcus faecium*, *Staphylococcus aureus*, *Klebsiella pneumoniae*, *Acinetobacter baumanni*, *Pseudomonas aeruginosa*, and *Enterobacter* spp.

#### 4.2.1. Aerobic Culture

Ten microliters of the homogenates were inoculated onto a 5% sheep’s blood agar plate and a MacConkey agar plate (Remel). The plates were then incubated in the 5% CO_2_ incubator for 48 h at 35 +/− 2°C. Growth was then observed at 18–24 and 48 h.

#### 4.2.2. Campylobacter Culture

Ten microliters of the homogenates were inoculated onto solid *Campylobacter* C.V.A. Medium (Remel). The plate was then placed in an EZ Pouch (B.D.) with a GasPakTM EZ Campy Pouch System (B.D.) and incubated at 42 + /−2°C. Growth was observed at 48 h.

#### 4.2.3. Salmonella Culture

For *Salmonella* spp. culture, 10 µL of the homogenates were inoculated onto Brilliant Green (B.G.) and XLT4 agar (Remel). Additionally, one mL of homogenate was added to 10 mL of tetrathionate broth. The plates and broth were incubated at 35 +/− 2°C. The B.G. agar was observed at 24 h, and the XLT4 was observed at 24 and 48 h. At 48 h, the tetrathionate broth was sub-cultured to a B.G. and XLT4 plate which were incubated and observed as described above.

#### 4.2.4. Bacterial Isolate Identification

Colonies observed on solid media were identified by Matrix-Assisted Laser Desorption Ionization Time-of-Flight mass spectrometry (MALDI-TOF MS). The MALDI-TOF MS detects bacterial proteins in whole-cell extracts resulting in a unique bacterial fingerprint that differentiates bacteria up to the species level [49]. For this procedure, each isolate was transferred onto the target plate (Bruker), and a 1 μL aliquot of matrix solution (a-Cyano-4-hydroxycinnamic acid, HCCA) was added. The MALDI-TOF spectrum for each isolate was automatically generated in the mass spectrometer. The spectra were collected in a mass range between 2000 and 20,000 m/s and were analyzed using the Biotyper^®^ software (Bruker, Bremen, Germany), which matched the spectrum of each sample against an extensive reference library containing the spectra of reference species. The results were interpreted on a 0–3 scale with a more precise match and reliable identification taking the highest value. Species-level identifications required a score of ≥2.0.

### 4.3. Antimicrobial Susceptibility Testing

Antimicrobial susceptibility testing followed the Clinical and Laboratory Standards Institute guidelines for veterinary medicine [50]. Minimum inhibitory concentrations (M.I.C.s) were determined using the Sensititre™ Vet Bovine BOPO7F plate (Thermo Scientific™, Waltham, MA, USA). The Sensititre™ Vet Bovine BOPO7F plate was used in accordance with the manufacturer’s instructions. Plates were read on the Sensititre™ Vizion™ (Thermo Scientific™) using the Sensititre™ SWIN™ software (Thermo Scientific™). Antimicrobials tested were ampicillin, ceftiofur, clindamycin, danofloxacin, enrofloxacin, florfenicol, gamithromycin, gentamycin, neomycin, penicillin, spectinomycin, sulphadimethoxine, tetracycline, tiamulin, tildipirosin, tilmicosin, trimethoprim/sulfamethoxazole, tulathromycin, and tylosin. The M.I.C.s were interpreted based on current CLSI guidelines for bacteria isolated from ruminants (CLSI Vet08, 4th Edition) by the Sensititre™ SWIN™ software version 3.3 (Thermo Scientific™).

### 4.4. Data Analyses

Antimicrobials that did not have an interpretive criterion for all or a specific pathogen were excluded from the data analyses. An isolate that was either resistant or intermediately susceptible to an antimicrobial was considered not susceptible as previously suggested [51,52]. Commercial statistical software (S.A.S., version 9.4, S.A.S. Institute Inc, Cary, NC, USA) was used to perform the descriptive analyses. Frequencies and proportions were used to summarize the data.

## 5. Conclusions

Antimicrobial-resistant bacterial species were identified in the freshwater mussel species *Lampsilis cardium* inhabiting one location of the Wildcat Creek watershed located directly downstream from Kokomo, Indiana. Freshwater mussels could be utilized as part of an AMR surveillance program. Further work is necessary to determine the upstream source of the bacterial species identified.

## Figures and Tables

**Figure 1 antibiotics-12-00728-f001:**
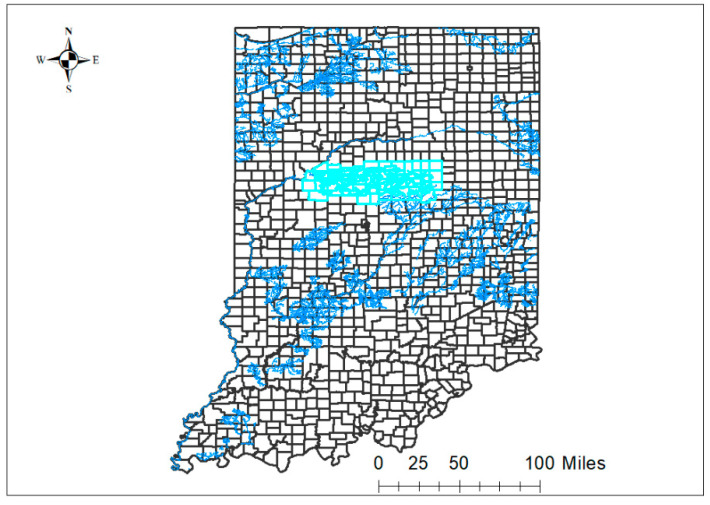
A map showing the location of the Wildcat Creek watershed (shaded sky-blue) in northcentral Indiana, United States. This map was created using ArcMap in ArcGIS version 10.5.1. The Indiana map shape file and associated metadata were downloaded from https://catalog.data.gov/dataset/tiger-line-shapefile-2017-state-indiana-current-county-subdivision-state-based/resource/e8b44c5a-4bb8-43c4-a2ce-ffc7c75924a0 accessed on 15 June 2020. (URL: https://www2.census.gov/geo/tiger/TIGER2017/COUSUB/tl_2017_18_cousub.zip).

**Figure 2 antibiotics-12-00728-f002:**
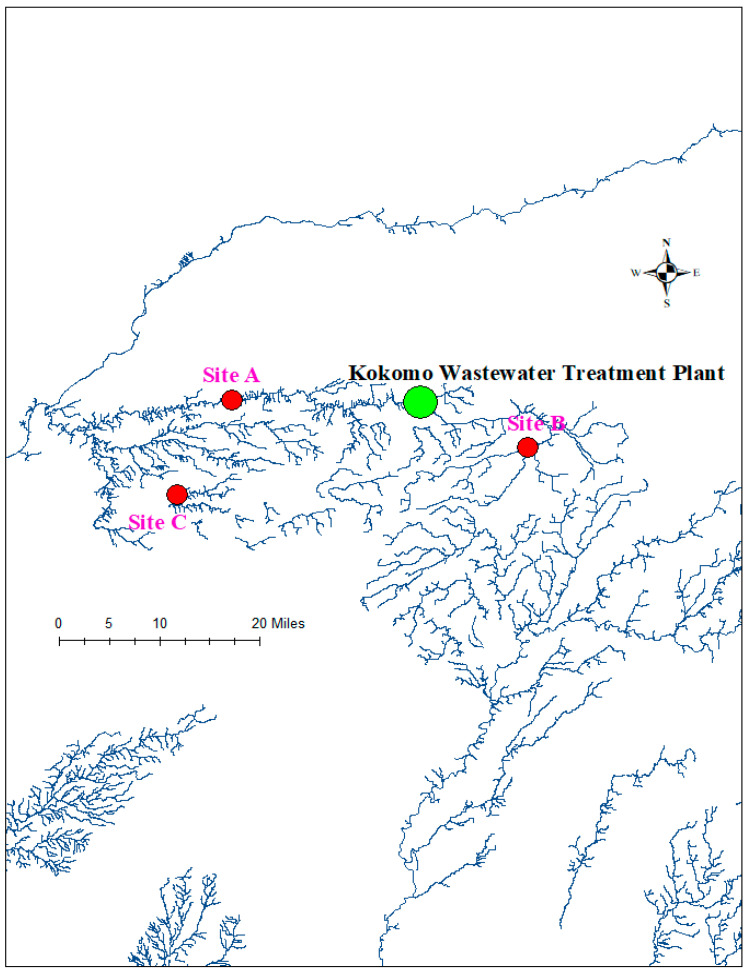
A map layer of impaired water bodies in Indiana. The locations of the three sampling sites within the Wildcat Creek watershed are represented by the letters A, B, and C. Site A is the Wildcat Creek (Carrol County), Site B is the Mud Creek (Tipton County), and Site C is the Kilmore Creek (Clinton County). This map was created using ArcMap in ArcGIS version 10.5.1. The map layer and associated metadata were downloaded from https://maps.indiana.edu/previewMaps/Hydrology/Water_Quality_Impaired_Waters_303d_2016.html on 15 June 2020. (Credits: Indiana Department of Environmental Management, U.S. Environmental Protection Agency).

**Table 1 antibiotics-12-00728-t001:** A summary of the antimicrobial susceptibility test results for bacteria isolated from the plain pocketbook freshwater mussels inhabiting the Wildcat Creek in north-central Indiana, United States.

Bacterial Organisms (Number of Isolates)	Antimicrobial Agents	Dilution Range (µg/mL)	M.I.C. (µg/mL)	Number of Susceptible Isolates
*Escherichia coli* (5)	Ampicillin	0.25–16	2–4 *	5
	Penicillin	0.12–8	>8	0
	Ceftiofur	0.25–8	≤0.25–0.5 *	5
	Gentamycin	1–16	<1	5
	Neomycin	4–32	≤4	5
	Spectinomycin	8–64	16–32 *	0
	Tilmicosin	2–16	>16	0
	Clindamycin	0.25–16	>16	0
	Sulphadimethoxine	256	≤256–>256 *	4
	Tetracycline	0.5–8	1–2 *	5
	Tiamulin	0.5–32	>32	0
	Florfenicol	0.25–8	2–8 *	1
*Enterobacter cloacae* (1)	Ampicillin	0.25–16	16	0
	Penicillin	0.12–8	>8	0
	Ceftiofur	0.25–8	1	1
	Gentamycin	1–16	≤1	1
	Neomycin	4–32	≤4	1
	Spectinomycin	8–64	16	0
	Tilmicosin	2–16	>16	0
	Clindamycin	0.25–16	>16	0
	Sulphadimethoxine	256	≤256	1
	Tetracycline	0.5–8	2	1
	Tiamulin	0.5–32	>32	0
	Florfenicol	0.25–8	4	0
*Klebsiella pneumoniae* (1)	Ampicillin	0.25–16	-	0
	Penicillin	0.12–8	>8	0
	Ceftiofur	0.25–8	1	1
	Gentamycin	1–16	≤1	1
	Neomycin	4–32	≤4	1
	Spectinomycin	8–64	16	0
	Tilmicosin	2–16	>16	0
	Clindamycin	0.25–16	>16	0
	Sulphadimethoxine	256	>256	0
	Tetracycline	0.5–8	4	1
	Tiamulin	0.5–32	>32	0
	Florfenicol	0.25–8	4	0
*Pseudomonas aeruginosa* (1)	Ampicillin	0.25–16	>16	0
	Penicillin	0.12–8	>8	0
	Ceftiofur	0.25–8	>8	0
	Gentamycin	1–16	≤1	1
	Neomycin	4–32	≤4	1
	Spectinomycin	8–64	>64	0
	Tilmicosin	2–16	>16	0
	Clindamycin	0.25–16	>16	0
	Sulphadimethoxine	256	>256	0
	Tiamulin	0.5–32	>32	0
	Florfenicol	0.25–8	>8	0

* = M.I.C. range. In this table, counts of susceptible isolates are presented instead of proportions because of the few numbers of isolates.

## Data Availability

The data for this study are available from the corresponding author upon reasonable request.

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
