# Peer review of "Antimicrobial Susceptibility of Bacteria Isolated from Freshwater Mussels in the Wildcat Creek Watershed, Indiana, United States"

_antibiotics, 2023, doi:10.3390/antibiotics12040728_

Round 1

Reviewer 1 Report

We commend your focus on mussels as an environmental reservoir for sampling to conduct antimicrobial resistance surveillance and taking a step beyond what is known in previous literature by conducting susceptibility testing. However, the present manuscript requires clarity of its methodology and correction to several references. Further, the current sample size is very small; conducting the further research that is proposed analysis in your discussion section would significantly strengthen the paper and should be completed prior to publication.

Some comments for consideration prior to resubmission are below:

1.    Introduction

a.    Mussels like other multicellular organisms have microbiomes. A comment about what is known already about this topic, such as that there is a difference in microbiome by species and environment is pertinent. If the microbiomes of the specific species included is unknown, please state as a limitation.

b.    Line 38: This sentence is phrased in a way such that Reference 4 is from WHO, or mentions WHO’s top ten threats to global health, which it does not.

2.    Results - Table 1

a.    Under E. coli and florfenicol: there was only one isolate reported for this susceptibility. Why is there a range of 3 dilutions?

b.    Please include breakpoints for reference. Readers may not have access to CLSI Vet08 4th edition.

3.    Discussion:

a.    Line 125: Please define medically important antibiotic.

b.    Lines 138 and 144: References 19 and 20 appear to be the same thing.

c.    Line 144: Pseudomonas aeruginosa is not intrinsically resistant to quinolones. Please re-check Table 2.4 from references 19 and 20, because this is incorrect.

d.    Line 148: discrepancy between the statement of inability to conduct fluoroquinolone susceptibility testing and statement in Methods section on Line 238-242 about testing susceptibilities for danofloxacin and enrofloxacin.

e.    Lines 155-158: Recommend increasing sample size and sampling wastewater as proposed to be included in this publication– as it stands the current sample size of 8 isolates is unlikely to make a significant impact compared to the current literature.

4.    Methods

a.    Lines 166-167: Why was each study site chosen?

b.    Lines 170-172: How long were mussels stored within the coolers on ice? Was the tissue of each mussel placed in its own sample bag to prevent contamination between mussels?

c.    Lines 172-174: Please explain the rationale for different transport methods of Asian clams compared to Plain Pocketbook and Fat Mucket mussels. Why were they held in a water bath for 18 hours?

Author Response

We commend your focus on mussels as an environmental reservoir for sampling to conduct antimicrobial resistance surveillance and taking a step beyond what is known in previous literature by conducting susceptibility testing. However, the present manuscript requires clarity of its methodology and correction to several references. Further, the current sample size is very small; conducting the further research that is proposed analysis in your discussion section would significantly strengthen the paper and should be completed prior to publication.

Response

We thank you so much for taking the time to review our paper. The feedback you have provided has greatly improved our manuscript. We have made clarifications in the methods and made corrections to the references. We believe that a sample size of 180 mussels is larger than the typical mussel sample sizes in previous studies. We acknowledge the fact that we report susceptibility of a few isolates. However, we also report that no bacteria were isolated from 120/180 mussels collected at two of the three sampling sites. We strongly believe that these negative results need to be published in scientific literature because this is new information that adds to the body of knowledge and could inform future AMR surveillance efforts as baseline data. In the revised manuscript, we discuss the likely importance of finding no bacteria in 120/180 mussels. Additionally, we believe that negative results should be published in an effort to reduce publication bias. Our belief is anchored on previous opinions on the importance of publishing negative results. (see Weintraub PG. The Importance of Publishing Negative Results. J Insect Sci. 2016 Oct 23;16(1):109. doi: 10.1093/jisesa/iew092. PMID: 27773876; PMCID: PMC5088693.)

Some comments for consideration prior to resubmission are below:

  1. Introduction
  2. Mussels like other multicellular organisms have microbiomes. A comment about what is known already about this topic, such as that there is a difference in microbiome by species and environment is pertinent. If the microbiomes of the specific species included is unknown, please state as a limitation.

Response: We thank you for this observation. We have commented on what is known about the microbiomes of mussels in our revised manuscript. Please see lines 66-68, and lines 153-166 in the revised manuscript.

  1. Line 38: This sentence is phrased in a way such that Reference 4 is from WHO, or mentions WHO’s top ten threats to global health, which it does not.

Response: We thank you for seeing this error. We have corrected this in the revised manuscript. Please see line 42 in the revised manuscript.

  1. Results - Table 1
  2. Under E. coli and florfenicol: there was only one isolate reported for this susceptibility. Why is there a range of 3 dilutions?

Response: There were five isolates tested and just one was susceptible. The MIC range is for all the isolates tested. We conformed to the standard and previously published method of reporting MICs for multiple isolates regardless of susceptibility status. We are happy to change this under the editor’s advice.

  1. Please include breakpoints for reference. Readers may not have access to CLSI Vet08 4th edition.

Response: We thank you for this suggestion. We have included the dilution range that we tested (see Table 1) and also clarified that the MICs were interpreted by the Sensititre™ SWIN™ software produced by Thermo Scientific (lines 315-316)

  1. Discussion:
  2. Line 125: Please define medically important antibiotic.

Response: We have defined a medically important antimicrobial. Please see lines 173-174. Thank you.

  1. Lines 138 and 144: References 19 and 20 appear to be the same thing.

Response: We have corrected this. Please see the reference list in the revised manuscript.

  1. Line 144: Pseudomonas aeruginosa is not intrinsically resistant to quinolones. Please re-check Table 2.4 from references 19 and 20, because this is incorrect.

Response: We checked the reference and the information in table 2.4 and table 3.2 conflict. Quinolones are listed in Table 3.2 of previously reference 19 (Antimicrobial Therapy in Veterinary Medicine Fifth Edition).  However, because of this conflicting information, we have removed quinolones from the list of intrinsically resistant classes in our manuscript. Please see line 205.

  1. Line 148: discrepancy between the statement of inability to conduct fluoroquinolone susceptibility testing and statement in Methods section on Line 238-242 about testing susceptibilities for danofloxacin and enrofloxacin.

Response: To clarify, these were tested, but their antimicrobial susceptibility profiles could not be determined using the Bovine BOPO7F AST Plate due to lack of interpretive criteria. The antimicrobial susceptibility reports indicated that these antimicrobials had no interpretive criteria. We mention how we handled this in lines 318 and 319.

  1. Lines 155-158: Recommend increasing sample size and sampling wastewater as proposed to be included in this publication– as it stands the current sample size of 8 isolates is unlikely to make a significant impact compared to the current literature.

Response: We thank you so much for this suggestion. However, we believe that the content we present in this manuscript is new scientific information that should be published. We believe that a sample size of 180 mussels is larger than the typical mussel sample sizes in previous studies. We acknowledge the fact that we report susceptibility of 8 isolates. However, we also report that no bacteria were isolated from 120/180 mussels collected at two of the three sampling sites. We strongly believe that this negative results need to be published in scientific literature because this is new information that adds to the body of knowledge and could inform future AMR surveillance efforts as baseline data. In the revised manuscript, we discuss the likely importance of finding no bacteria in 120/180 mussels. Additionally, we believe that negative results should be published in an effort to reduce publication bias. At the moment, we have not yet secured funding for conducting our planned follow up studies. We believe that publishing findings from this pilot study will give us leverage for securing funding for our planned follow up studies.

  1. Methods
  2. Lines 166-167: Why was each study site chosen?

Response: As noted in the manuscript (lines 226-227), these samples were collected as part of a larger study looking at mussel populations in the Wildcat Creek. This pilot study was conducted through convenience sampling at the sites identified as ecologically important for the larger study.

  1. Lines 170-172: How long were mussels stored within the coolers on ice? Was the tissue of each mussel placed in its own sample bag to prevent contamination between mussels?

Response: This information has been clarified in lines 231-235.

  1. Lines 172-174: Please explain the rationale for different transport methods of Asian clams compared to Plain Pocketbook and Fat Mucket mussels. Why were they held in a water bath for 18 hours?

Response: This was due to weather conditions in the field. The Asian Clams were the last collected in that site and were help overnight in a water bath until we could process them. This has been added in lines 235-239.

Reviewer 2 Report

The introduction section needs to be refined with reference to the water microbiology. Is there any ecological relationship between freshwater mussels and bacteria?

What are the bacterial flora characteristics of freshwater mussels?

How the study of antimicrobial resistance is significant

What are the most common reasons why bacteria become resistant to antibiotics? Is the resistant pattern found in a sample of water other than the muscle?

The methodology is poorly described and needs to be significantly improved.

In the section called "Results and Discussion," MALDI-TOF is only mentioned as a way to find out what kind of bacteria there are; its method and how the results should be interpreted are not mentioned.

The focus of the study is on bacteria that are resistant to antibiotics. A significant description of a resistant gene for any identified bacteria would be significant.

There is poor article use in the languages of the manuscript, which must be improved.

Author Response

The introduction section needs to be refined with reference to the water microbiology. Is there any ecological relationship between freshwater mussels and bacteria?

Response: We thank you so much for reviewing our paper and providing feedback. Your review feedback has greatly improved our manuscript. We have extensively refined the introduction and discuss relevant aspects of surface water microbiology and the ecological relationship between freshwater mussels and bacteria. Please see lines 46-59.

What are the bacterial flora characteristics of freshwater mussels?

Response: We thank you for asking this question. We address this in the revised manuscript in lines 66-68, and 153-166.

How the study of antimicrobial resistance is significant

Response: We have addressed this in the introduction. Please see lines 38-40.

What are the most common reasons why bacteria become resistant to antibiotics? Is the resistant pattern found in a sample of water other than the muscle?

Response: We have addressed these questions in lines 38-40 and 79-81 and provided appropriate references.

The methodology is poorly described and needs to be significantly improved.

Response: We thank you for this comment. We have revised the methods section to address this concern.

In the section called "Results and Discussion," MALDI-TOF is only mentioned as a way to find out what kind of bacteria there are; its method and how the results should be interpreted are not mentioned.

Response: We thank you for this comment. We have revised the methods section to address this concern. Please see lines 296-307.

The focus of the study is on bacteria that are resistant to antibiotics. A significant description of a resistant gene for any identified bacteria would be significant.

Response: We have added a discussion of resistance genes. Please see lines 169 to 206. We thank you for this suggestion.

 There is poor article use in the languages of the manuscript, which must be improved.

Response: We have edited the paper to improve the language used. We thank you for this comment.

Round 2

Reviewer 1 Report

None

Reviewer 2 Report

All the comments raised have been addressed and satisfied.